# Drought Monitoring Based on Remote Sensing in a Grain-Producing Region in the Cerrado–Amazon Transition, Brazil

**Mairon Ânderson Cordeiro Correa de Carvalho [1], Eduardo Morgan Uliana [2],
Demetrius David da Silva [3], Uilson Ricardo Venâncio Aires [4,*],
Camila Aparecida da Silva Martins [5], Marionei Fomaca de Sousa Junior [6],
Ibraim Fantin da Cruz [7] and Múcio André dos Santos Alves Mendes [2]**

[1] Programa de Pós-Graduação em Recursos Hídricos, Universidade Federal de Mato Grosso, Cuiabá 78060-900, Brazil; maironcdecarvalho@gmail.com

[2] Instituto de Ciências Agrárias e Ambientais, Universidade Federal de Mato Grosso, Sinop 78557-267, Brazil; morganuliana@gmail.com (E.M.U.); mucioandre@gmail.com (M.A.d.S.A.M.)

[3] Departamento de Engenharia Agrícola, Universidade Federal de Viçosa, Viçosa 36570-900, Brazil; demetrius@ufv.br

[4] Programa de Pós-Graduação em Engenharia Agrícola, Universidade Federal de Viçosa, Viçosa 36570-900, Brazil;

[5] Departamento de Engenharia Rural, Universidade Federal do Espírito Santo, Alegre 29500-000, Brazil; camila.cca@hotmail.com

[6] Programa de Pós-Graduação em Sensoriamento Remoto, Instituto Nacional de Pesquisas Espaciais, São José dos Campos 12227-010, Brazil; mariofomacajr@gmail.com

[7] Departamento de Engenharia Sanitária e Ambiental, Universidade Federal de Mato Grosso, Cuiabá 78060-900, Brazil; ibraimfantin@gmail.com

**\*** Correspondence: uilson.aires@ufv.br; Tel.: +55-035-99161-2562

**Abstract:** Drought is a natural disaster that affects a country's economy and food security. The monitoring of droughts assists in planning assertive actions to mitigate the resulting environmental and economic impacts. This work aimed to evaluate the performance of the standardized precipitation index (SPI) using rainfall data estimated by orbital remote sensing in the monitoring of meteorological drought in the Cerrado–Amazon transition region, Brazil. Historical series from 34 rain gauge stations, in addition to indirect measurements of monthly precipitation obtained by remote sensing using the products CHIRPS-2.0, PERSIANN-CDR, PERSIANN-CCS, PERSIANN, GPM-3IMERGMv6, and GPM-3IMERGDLv6, were used in this study. Drought events detected by SPI were related to a reduction in soybean production. The SPI calculated from the historical rain series estimated by remote sensing allowed monitoring droughts, enabling a high detailing of the spatial variability of droughts in the region, mainly during the soybean development cycle. Indirect precipitation measures associated with SPI that have adequate performance for detecting droughts in the study region were PERSIANN-CCS (January), CHIRPS-2.0 (February and November), and GPM-3IMERGMv6 (March, September, and December). The SPI and the use of precipitation data estimated by remote sensing are effective for characterizing and monitoring meteorological drought in the study region.

**Keywords:** agricultural planning; soybean; climate risk; natural disaster; water resource management

## 1. Introduction

Drought is one of the most challenging and complex natural disasters, with negative impacts on the economy and environment [1–6]. Agriculture activities are highly impacted by drought, putting food production at risk, especially in regions with a predominance of rainfed agriculture. Losses due to droughts in regions whose economy is based on agribusiness affect the entire sector, such as services suppliers, machinery and agricultural implements, and fertilizers and agricultural defensives [2,4,7–11].

As the drought persists over time, other problems can directly affect the population, such as water supply, conflicts of water use, reduced electrical energy generation, and risks to public health due to low water quality [7,9,11–13]. Brazil is highly dependent on hydraulic energy and severe droughts have already caused energy rationing and an increase in the prices of electricity.

Droughts reduce biodiversity, leading to vegetation and animal death, increase the risk of forest fires, and alter soil physical properties, with a consequent change in runoff [8,10,14,15]. The work in [16] observed that droughts result in a reduction in the vegetative development rate and biomass accumulation, with significant consequences for carbon sequestration. [17,18] pointed out that droughts in the Amazon rainforest increased deforestation due to forest fires, with an increase in forest fragmentation and greenhouse gas emission.

With climate change, the events of droughts have shown a tendency to increase in its intensity and occurrence [2,7,19–22]. Thus, the development of methodologies for monitoring drought events is essential for proper planning of water use and decision making to prevent and mitigate drought impacts.

According to [20,23], the monitoring of drought can be obtained by applying drought indices to characterize its severity and impacts. There are several drought indices, with different input data specific to the type of drought aimed to be studied (meteorological, agricultural, or hydrological) [24,25].

Among these drought indices, [26] highlighted that the standardized precipitation index (SPI) is the most used and recommended by the World Meteorological Organization (WMO) as the standard drought index. The great advantage of SPI relative to the other indices is that it needs only the rainfall data as an input variable [27].

In many developing countries the monitoring of rainfall does not continue over time and the spatial distribution of the rain gauge stations is not enough to precisely estimate the drought on a large scale. In this context, the rainfall data estimated by orbital remote sensing is a powerful tool with great potential for monitoring droughts, as it provides a synoptic and repetitive view of the surface and allows a consistent, and systematic collection of data [26]. However, it is necessary to evaluate the application of the estimated rainfall products to obtain the drought indices [28].

Thus, this paper tested different products of rainfall data estimated by orbital remote sensing, which can be an alternative for regions that have a lack of rain gauge stations. This work provides information that can be used by decision-makers to monitor the drought events aimed to minimize the impacts of this natural phenomenon on agricultural activities. In addition, this information can help the government to adopt policies to mitigate this problem, guaranteeing the sustainability of agricultural activities in the study area. In this context, this work aimed to evaluate the performance of the standardized precipitation index (SPI) using rainfall data estimated by orbital remote sensing in the monitoring of meteorological drought in the Cerrado–Amazon transition region, Brazil.

## 2. Materials and Methods

### 2.1. Study Area and Database

The study area was the north–central region of the State of Mato Grosso, Brazil (Figure 1), located in the transition between the Cerrado and the Brazilian Amazon. It is a region that stands out for grain production, with a soybean and corn production in the 2019/2020 growing season of 11,867 and 15,079 thousand tons, respectively, according to the Instituto Mato-grossense de Economia Agropecuária [29,30]. These values correspond to approximately 34% and 45% of the soybean and

corn production, respectively, in the entire State of Mato Grosso in the 2019/2020 growing season. Most of the region's grain production is carried out under rainfed conditions, which makes it highly vulnerable to drought.

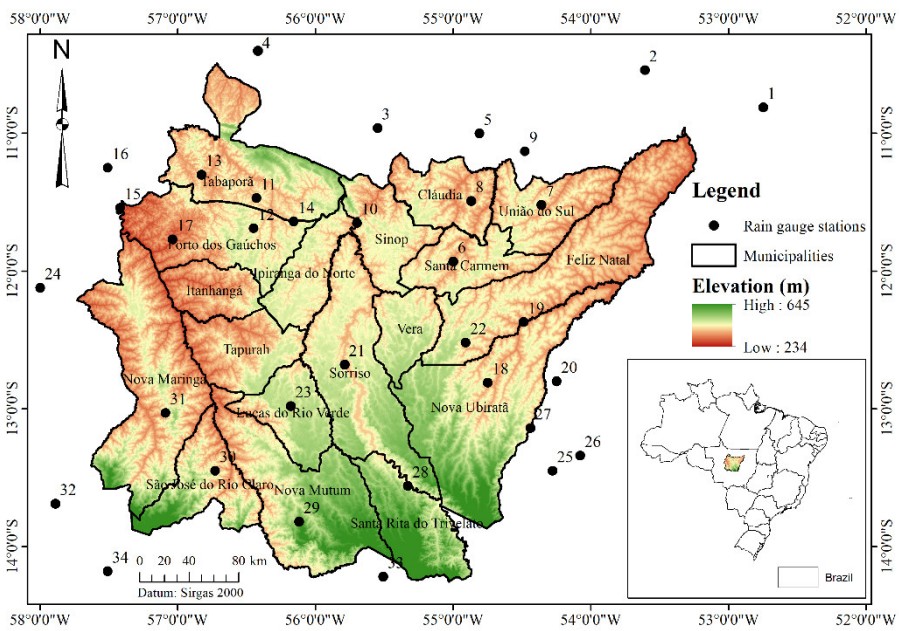

**Figure 1.** Map of the study area, location of the rain gauge stations belonging to the hydrometeorological network of the National Water Agency, and municipalities included in the north–central region of the State of Mato Grosso, Brazil.

The normal soybean sowing season in the region occurs in the second half of October and November [31], while the harvest is carried out from February to March. The second crop corn is sown right after the soybean harvest, with a sowing time limit until 15 March [32]. The second crop corn is usually harvested between June and July. These periods refer to the normal sowing time and disregard the development variability between cultivars. According to the Köppen classification, the regional climate is Aw, i.e., a tropical warm and humid climate with two well-defined seasons: drought from May to September and rainy from October to April [33].

Thirty-four historical rain series recorded in rain gauge stations located in and around the north–central region of the State of Mato Grosso and belonging to the hydrometeorological network of the Agencia Nancional de Águas (ANA) were used to carry out this study. The location and information related to the rain gauge stations are shown in Figure 1 and Appendix Table A1. The historical precipitation data series was obtained through the Sistema de Informações Hidrológicas (HidroWeb) [34].

Figure 2 shows the behavior of the probable average monthly precipitation for the study region. In this case, the probable precipitation refers to the height of rain that has a probability of occurrence higher than or equal to 75%. The probable annual precipitation (PAP) of the region has an average value of 1622 mm. The monthly rainfall probability was obtained following the methodology proposed by [35]. Two well-defined seasons are found in the north–central region of the State of Mato Grosso, i.e., a rainy season from October to April and a dry season between May and September. The months from April to October are considered the transition between both seasons. January has the highest probable amount of rain, but it presents a value very close to that of December.

June, July, and August show a zero probable average precipitation. The lowest rain values occur between May and September, a period in which agricultural production requires supplementary irrigation since the possibility of replacing water in the soil due to a natural rain event is very small. That is, it is not possible to produce in this period with rainfed agriculture, which is usually practiced during the other months of the year. The soybean sanitary break is practiced in the region from 15

June to 15 September, i.e., the presence of live soybean plants on the properties is prohibited [36]. The objective of the sanitary break is to reduce the occurrence of diseases that affect soybean cultivation, especially Asian rust. Thus, the main soybean development period is between October and March.

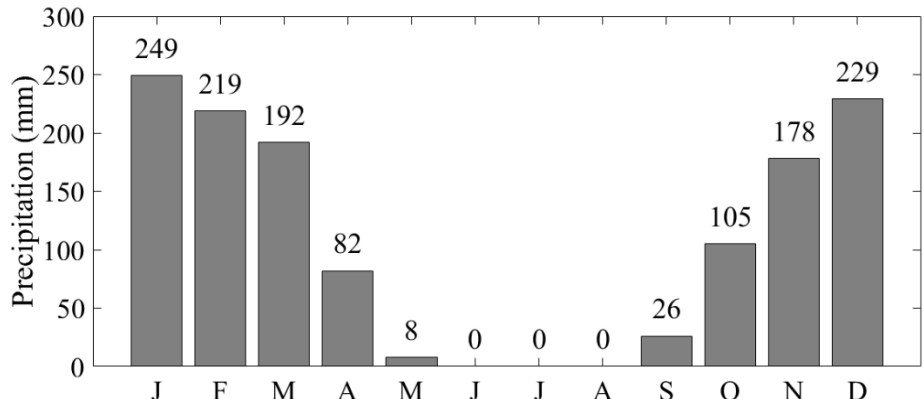

**Figure 2.** Probable average monthly precipitation (mm) estimated at the 75% probability level for the north–central region of the State of Mato Grosso, Brazil.

### 2.2. Drought Detection with SPI Obtained Based on Data from Rain Gauge Stations

The drought was characterized using the SPI (SPI) [37] because it is the main index recommended by the WMO, as mentioned by [20]. The historical precipitation series of 34 rain gauge stations (Appendix Table A1) were used as the input variable for calculating SPI, organized for the monthly time scale (SPI-1). The adoption of SPI on a monthly scale was performed because periods longer would cause difficulties for the drought analysis and its relationship with the phenological development phases of the region's crops, especially soybeans. The monthly SPI represents the meteorological drought that occurred prior to the agricultural drought; however, there is a direct relationship between these two types of drought. However, June, July, and August were disregarded because these months had a zero probable average precipitation under normal conditions (Figure 2).

The cumulative probability of occurrence of rain P(x) was determined using the theoretical gamma probability model after organizing the historical precipitation series on the monthly scale [36]. The probability density function of this model is shown in Equation (1).

$$f(x) = \frac{1}{\beta^{\gamma}\Gamma(\gamma)} x^{\gamma-1} e^{\frac{-x}{\beta}} \; ; \text{ para } \beta > 0 \text{ e } \gamma > 0 \tag{1}$$

where x is the monthly precipitation for a given time scale (mm), $\gamma$ is the shape parameter, $\beta$ is the scale parameter, and $\Gamma$ is the gamma function. This function was obtained by Equation (2).

$$\Gamma(\gamma) = \int_{0}^{\infty} x^{\gamma-1} e^{-x} dx \, , \; \gamma > 0 \tag{2}$$

where x is the monthly precipitation for a given time scale (mm) and $\gamma$ is the shape parameter.

The shape and scale parameters of the gamma distribution were obtained using the maximum likelihood method from historical rain data. More information on the maximum likelihood method can be obtained in [36].

The confirmation of the adherence of the gamma probability distribution to the rain data series was verified by the Kolmogorov–Smirnov (KS) adherence test at the 10% significance level, according to [38]. The choice of a higher significance level makes the KS test more rigorous to verify the adherence of a given probability distribution to the data series. For this reason, the KS test was applied at the 10% probability level.

After obtaining the parameters and verifying the adherence of the gamma distribution to the rain data series in the north–central region, the cumulative probability (P) of precipitation occurrence was calculated through the numerical integration of Equation (1), as shown in Equation (3).

$$P(x)= \int_0^x \frac{1}{\beta^\gamma \Gamma(\gamma)} x^{\gamma-1} e^{\frac{-x}{\beta}}\, dx \tag{3}$$

where $P(x)$ is the cumulative probability of monthly precipitation, $x$ is precipitation (mm), $\gamma$ is the shape parameter, $\beta$ is the scale parameter, and $\Gamma$ is the gamma function.

The cumulative probability value was corrected using Equations (4) and (5) because the gamma distribution is undefined for values equal to zero, that is, the absence of rain.

$$P(x)=q+(1-q).p(x) \tag{4}$$

$$q= \frac{m}{n} \tag{5}$$

where $P(x)$ is the corrected probability of precipitation occurrence, $q$ is the probability of zero precipitation, $m$ is the number of rain values equal to zero, and $n$ is the total number of values in the sample.

The probability value $P(x)$ was transformed into a normal variable using Equations (6) and (7) proposed by [39]. This transformation resulted in the SPI value.

$$SPI = -\left(t - \frac{c_0 + c_1 t + c_2 t^2}{1 + d_1 t + d_2 t^2 + d_3 t^3}\right),\ 0 < P(x) \le 0.5 \tag{6}$$

$$SPI = +\left(t - \frac{c_0 + c_1 t + c_2 t^2}{1 + d_1 t + d_2 t^2 + d_3 t^3}\right),\ 0.5 < P(x) \le 1 \tag{7}$$

where $c_0$ = 2.515517, $c_1$ = 0.802853, $c_2$ = 0.010328, $d_1$ = 1.432788, $d_2$ = 0.189269, and $d_3$ = 0.001308. Moreover,

$$t= \sqrt{\ln\left(\frac{1}{(P(x))^2}\right)}\ , para\ 0 < P(x) \le 0.5 \tag{8}$$

$$t= \sqrt{\ln\left(\frac{1}{(1-P(x))^2}\right)}\ , para\ 0.5 < P(x) \le 1 \tag{9}$$

The SPI value was classified according to the limits established in Table 1, modified from [27]. In this case, drought occurs when the SPI value is lower than or equal to −1. The severity of drought or humidity is classified by considering the magnitude of SPI.

**Table 1.** Classification of rain anomaly according to the standardized precipitation index (SPI) values obtained using Equation (6) or (7) (modified from [27]).

| Class | SPI Value | Simplified Classification |
|---|---|---|
| Extremely wet (EW) | SPI ≥ 1.5 | |
| Moderately wet (MW) | 1 ≤ SPI < 1.5 | Without drought |
| Normal precipitation (NP) | 1 < SPI < −1 | |
| Moderately dry (MD) | −1.5 < SPI ≤ −1 | With drought |
| Extremely dry (ED) | SPI ≤ −1.5 | |

The Mann–Kendall test at a 5% significance was used to verify the existence of a significant trend in the historical series of monthly SPI.

### 2.3. SPI Estimation Using Orbital Remote Sensing

This study used the precipitation data recorded on the surface and those estimated by remote sensing. Table 2 shows the precipitation products from remote sensing used in this study, their spatial resolution, the extent of the historical series, and the electronic address where the images were downloaded. The information related to the databases is detailed in these electronic addresses.

The time scale of all products (Table 2) is monthly, except for GPM-3IMERGDLv6, which is daily, and the total precipitated accumulation was performed for each month.

**Table 2.** Information related to rain estimations by remote sensing used in this study.

| Product | Spatial Resolution | Period | Database Address |
|---------|-------------------|--------|------------------|
| CHIRPS-2.0 | 0.05° | 1981–2020 | https://www.chc.ucsb.edu/data/chirps |
| PERSIANN-CDR | 0.25° | 1983–2020 | |
| PERSIANN-CCS | 0.04° | 2003–2020 | https://chrsdata.eng.uci.edu/ |
| PERSIANN | 0.25° | 2000–2020 | |
| GPM-3IMERGMv6 | 0.10° | 2000–2020 | |
| GPM-3IMERGDLv6 | 0.10° | 2000–2020 | https://giovanni.gsfc.nasa.gov/giovanni/ |

The SPI was determined using a routine developed in the software Matlab R2016b (OPENCADD, São Paulo, Brazil) for reading the images and forming a historical series of monthly rain for each pixel of the products described in Table 2. The same statistical procedure described above was performed to determine SPI-1 with the historical series obtained pixel by pixel. Thus, the monthly precipitation images for each year of the historical series were transformed into SPI-1 images.

A preliminary analysis showed a divergence between the categories of SPI (Table 1) obtained with data from rain gauges (Appendix Table A1) compared to satellite images (Table 2). The results of the satellite data did not allow adequate characterization of the severity of drought or humidity according to the classification shown in Table 1, but they presented a satisfactory agreement in indicating the existence or not of the drought event. Thus, two SPI classes were considered at this stage: With drought (SPI ≤ −1) and Without drought (SPI > −1).

The validation of the accuracy of SPI, obtained from satellite images (Table 2), in detecting drought events was carried out by calculating the global accuracy, confidence interval of the global accuracy, tau index ($\tau$) [40], and kappa index [41] and performing the McNemar test [42] at the 5% significance level. For this, an error matrix was constructed considering as reference the SPI values obtained for the rain gauges and the classes WITH drought and Without drought (Table 1). The quality of the drought classification associated with the values of the kappa and tau ($\tau$) indices was performed according to the criteria shown in Table 3.

**Table 3.** Quality of drought classification associated with kappa and tau ($\tau$) statistics.

| Kappa or Tau ($\tau$) | Drought Map Quality |
|-----------------------|---------------------|
| <0.00 | Very poor |
| 0.00–0.20 | Poor |
| 0.20–0.40 | Moderate |
| 0.40–0.60 | Good |
| 0.60–0.80 | Very good |
| 0.80–1.00 | Excellent |

Source: adapted from [41].

The global accuracy ($P_0$), and the tau ($\tau$), and Kappa (K) indices were obtained according to [43]. After the construction of the error matrix, the general accuracy was calculated by Equation (10).

$$P_0 = \frac{\sum_{i=j}^{M} N_{ij}}{N} =$$
(10)

where: N is the total number of samples; $N_{ij}$, whenever i = j, represents the number of correct classifications; M the number of classes, in this work equal to 2 ("with drought"; "without drought").

Then the Kappa index was determined using Equations (11) and (12).

$$K= \frac{P_0 - P_c}{1 - P_c} \qquad (11)$$

$$P_c= \frac{\sum_{i=1}^{M} N_{i+}N_{+j}}{N^2} \qquad (12)$$

where $P_0$ is the global accuracy; $P_c$ the proportion of units that agree by chance; m, the number of classes present in the error matrix; $N_{i+}$ and $N_{+j}$ the marginal totals of row i and column j, respectively; and N, the total number of samples.

The calculation of the tau index was performed with Equation (13).

$$\tau = \frac{P_o - \frac{1}{M}}{1 - \frac{1}{M}} \qquad (13)$$

where $P_0$ is the global accuracy and M the number of classes, in this work equal to 2 ("with drought"; "without drought").

The McNemar test was used in this study, as recommended by [44]. The null hypothesis ($H_0$) of the statistical test states that there is no significant difference between classifiers. Therefore, the drought detection performed with the rain gauge and satellite data does not differ statistically if the null hypothesis is not rejected. Only the products shown in Table 2 were considered able for detecting drought, as they showed an overall accuracy higher than or equal to 0.7 (70%), kappa and tau values higher than 0.21, and the non-rejection of the $H_0$ hypothesis of the McNemar test.

McNemar's test is based on a chi-square statistic, calculated using Equation (14).

$$X^2= \frac{(|N_{12} - N_{21}| - 1)^2}{N_{12}+N_{21}} \qquad (14)$$

where: $N_{12}$ represents the number of misclassified of the first classifier and the right classification of the second classifier; $N_{21}$ the number of times the first classifier estimated drought correctly and the number of misclassified of the second classifier.

## 3. Results and Discussion

### 3.1. Drought Detection with SPI Obtained Based on Data from Rain Gauge Stations

Figure 3 shows the mean values of the monthly standardized precipitation index (SPI-1) of the north–central region of the State of Mato Grosso for the period from 1985 to 2017. June, July, and August were disregarded from the analysis because they were in the normally dry period and presented a rain value equal to zero, as shown in Figure 2.

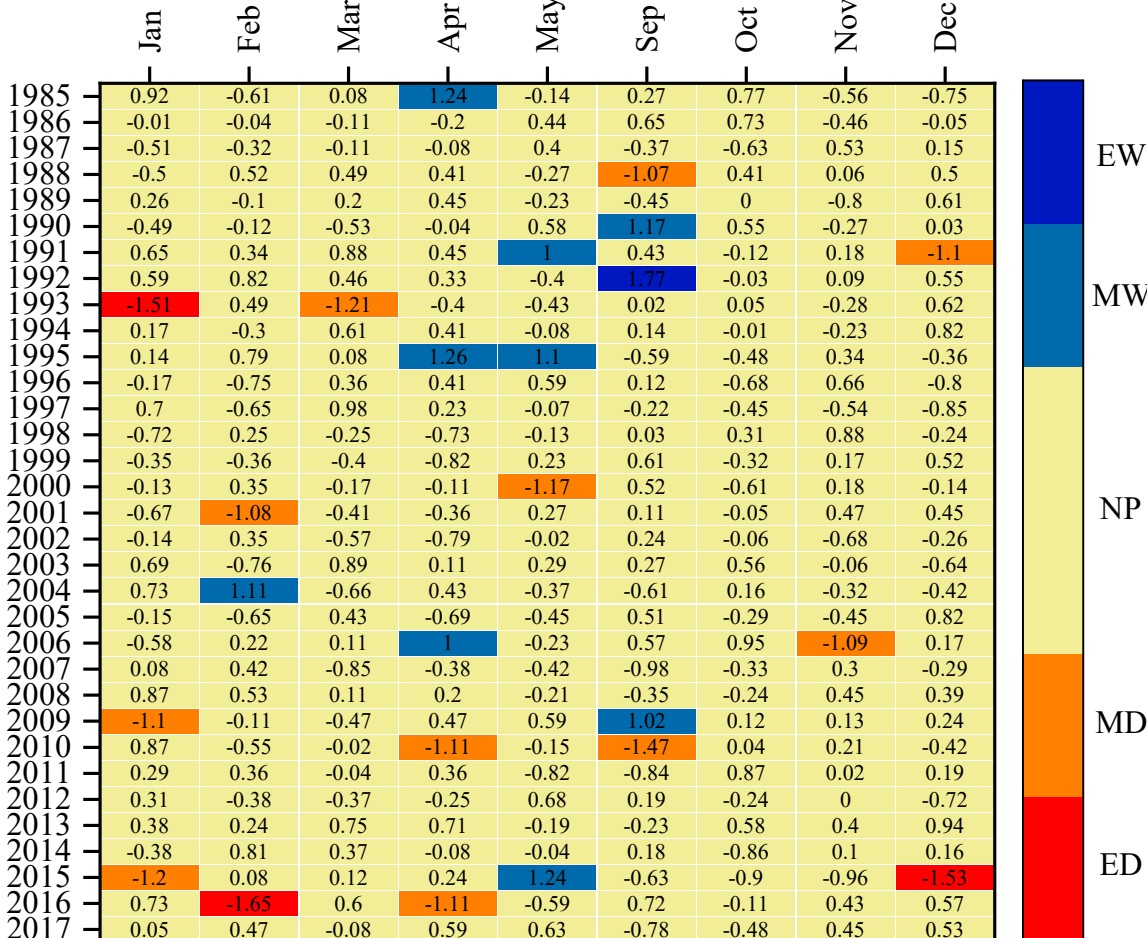

**Figure 3.** Annual average monthly SPI based on data from 34 rain gauge stations located in the north–central region of the State of Mato Grosso from 1985 to 2017. EW: extremely wet; MW: moderately wet; NP: normal precipitation; MD: moderately dry; ED: extremely dry.

The analysis of the average monthly SPI values (Figure 3) showed that more than 91.9% of the monthly accumulated precipitation occurring in the study region can be categorized by SPI as normal precipitation (NP), while 4.7% was classified as extremely dry (ED) and moderately dry (MD) and 3.4% as moderately wet (MW) and extremely wet (EW).

Figure 3 shows that the extreme drought (ED) occurred in three months of three distinct years of the historical series, that is, January 1993, December 2015, and February 2016. These events occurred exactly during the soybean development period in the study region. The 2015/2016 soybean-growing season was one of the most affected due to drought. An 11% reduction in production was observed in this growing season compared to the previous growing season (2014/2015), even with an increase in planted area by 1% (Figure 4). There was a production increase in 2015/2016, even with drought greater than 2012/2013, due to an increase in planted area. The location of the study region is in the north and central west of Brazil, where the exploitation of land by primary production is still expanding, with the removal of vegetation cover for the implementation of agriculture.

Agricultural production in the region is conducted under rainfed conditions and the sensitivity of SPI for detecting droughts and its impact on the grain-growing season in the region is evident. The study in [20] used SPI to monitor droughts in the north and northwest regions of the State of Rio de Janeiro, Brazil, and concluded that the monthly SPI is efficient for detecting extreme droughts. The results of this research corroborate those obtained for the north–central region of the State of Mato Grosso regarding SPI, but the data of Figures 3 and 4 also show that this index is related to a decrease in soybean production, such as that which occurred in the 2015/2016 growing season (Figure 4).

The study in [45] used the monthly SPI to monitor droughts in the State of Espírito Santo, Brazil, and found that its values could detect droughts, leading to a reduction in coffee production. This result and that obtained for the region under study demonstrate that the monthly SPI can detect droughts and their consequences for different crops.

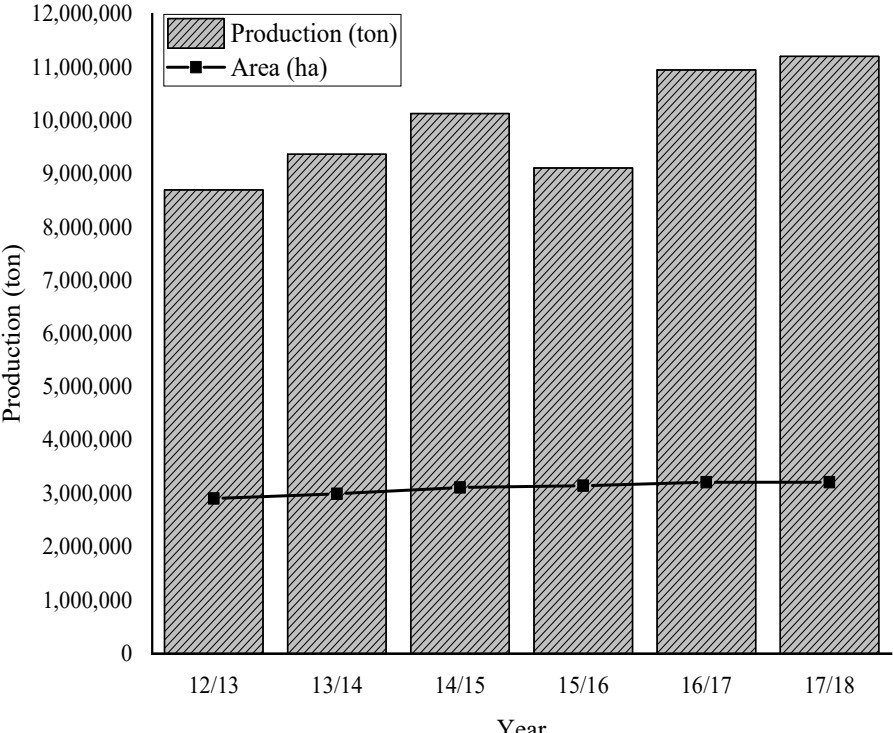

**Figure 4.** Production per growing season and the annual area used for soybean sowing in the north–central region of the State Mato Grosso, Brazil. Source: [46].

The trend analysis of the monthly SPI values (Table 4) by the Mann–Kendall hypothesis test indicated that the series has stationarity, that is, the SPI values are invariant regarding the chronology of their occurrences, except for random fluctuations. Thus, there is no trend to increase or decrease dry and humid events in the north–central region of the State of Mato Grosso for the analyzed period.

**Table 4.** Results of the Mann–Kendall trend test at the 5% significance level for the monthly SPI data shown in Figure 3.

| Result | January | February | March | April | May | September | October | November | December |
|---|---|---|---|---|---|---|---|---|---|
| Z | 0.418 | 0.201 | −0.124 | −0.759 | −0.232 | −0.961 | −1.224 | 1.038 | 0.294 |
| *p*-value | 0.676 | 0.840 | 0.901 | 0.448 | 0.816 | 0.337 | 0.221 | 0.299 | 0.768 |

*3.2. Drought Detection with SPI Obtained Based on Orbital Remote Sensing Data*

Table 5 shows the validation of different rain estimations by remote sensing to detect droughts in the study region using SPI. The simplified SPI scale is shown in Table 1 with the classes With drought and Without drought. The results in bold indicate an adequate performance of the products for detecting droughts according to criteria established in the methodology. June, July, and August were also disregarded from the analysis, as these months are usually dry in the region under study.

**Table 5.** Validation of drought estimations based on remote sensing data using the simplified SPI classification.

| Month | Product | Global Accuracy | CI-Accuracy | Tau | Kappa | Kappa CLASSIFICATION | *p*-Value—McNemar |
|---|---|---|---|---|---|---|---|
| January | **CHIRPS-2.0** | **0.86** | **0.82–0.89** | **0.72** | **0.50** | **Good** | **0.228** |
| | PERSIANN-CDR | 0.84 | 0.80–0.87 | 0.68 | 0.41 | Good | 0.018 |
| | **PERSIANN-CCS** | **0.81** | **0.76–0.85** | **0.62** | **0.31** | **Moderate** | **0.791** |
| | **PERSIANN** | **0.85** | **0.81–0.88** | **0.70** | **0.43** | **Good** | **0.161** |
| | **GPM-3IMERGMv6** | **0.86** | **0.81–0.89** | **0.72** | **0.46** | **Good** | **0.112** |
| | **GPM-3IMERGDLv6** | **0.86** | **0.83–0.88** | **0.72** | **0.50** | **Good** | **0.097** |
| February | **CHIRPS-2.0** | **0.85** | **0.82–0.88** | **0.7** | **0.50** | **Good** | **0.723** |
| | **PERSIANN-CDR** | **0.82** | **0.79–0.85** | **0.64** | **0.37** | **Moderate** | **0.299** |
| | **PERSIANN-CCS** | **0.84** | **0.79–0.88** | **0.68** | **0.40** | **Good** | **0.567** |
| | **PERSIANN** | **0.79** | **0.74–0.83** | **0.58** | **0.24** | **Moderate** | **0.194** |
| | **GPM-3IMERGMv6** | **0.84** | **0.79–0.87** | **0.68** | **0.47** | **Good** | **0.504** |
| | **GPM-3IMERGDLv6** | **0.84** | **0.79–0.88** | **0.68** | **0.44** | **Good** | **0.418** |
| March | **CHIRPS-2.0** | **0.82** | **0.78–0.85** | **0.64** | **0.31** | **Moderate** | **0.668** |
| | **PERSIANN-CDR** | **0.80** | **0.76–0.83** | **0.60** | **0.26** | **Moderate** | **0.121** |
| | **PERSIANN-CCS** | **0.82** | **0.77–0.86** | **0.64** | **0.31** | **Moderate** | **0.418** |
| | PERSIANN | 0.72 | 0.66–0.76 | 0.44 | 0.04 | Poor | 0.036 |
| | **GPM-3IMERGMv6** | **0.79** | **0.75–0.84** | **0.58** | **0.30** | **Moderate** | **0.228** |
| | **GPM-3IMERGDLv6** | **0.79** | **0.74–0.83** | **0.58** | **0.27** | **Moderate** | **0.342** |
| April | CHIRPS-2.0 | 0.83 | 0.79–0.86 | 0.66 | 0.42 | Good | 0.030 |
| | PERSIANN-CDR | 0.80 | 0.76–0.83 | 0.6 | 0.36 | Moderate | 0.001 |
| | **PERSIANN-CCS** | **0.84** | **0.79–0.88** | **0.68** | **0.41** | **Good** | **0.999** |
| | **PERSIANN** | **0.82** | **0.78–0.86** | **0.64** | **0.41** | **Good** | **0.093** |
| | **GPM-3IMERGMv6** | **0.83** | **0.78–0.86** | **0.66** | **0.40** | **Good** | **0.434** |
| | **GPM-3IMERGDLv6** | **0.81** | **0.76–0.85** | **0.62** | **0.35** | **Moderate** | **0.175** |
| May | **CHIRPS-2.0** | **0.83** | **0.80–0.87** | **0.66** | **0.38** | **Moderate** | **0.075** |
| | PERSIANN-CDR | 0.78 | 0.74–0.81 | 0.56 | 0.18 | Poor | 0.021 |
| | PERSIANN-CCS | 0.78 | 0.72–0.84 | 0.56 | 0.16 | Poor | 0.033 |
| | PERSIANN | 0.81 | 0.76–0.85 | 0.62 | 0.17 | Poor | 0.048 |
| | GPM-3IMERGMv6 | 0.82 | 0.77–0.86 | 0.64 | 0.28 | Moderate | 0.007 |
| | GPM-3IMERGDLv6 | 0.80 | 0.75–0.84 | 0.6 | 0.27 | Moderate | $1.17 \times 10^{-4}$ |
| September | **CHIRPS-2.0** | **0.88** | **0.85–0.91** | **0.76** | **0.55** | **Good** | **0.602** |
| | **PERSIANN-CDR** | **0.87** | **0.84–0.90** | **0.74** | **0.52** | **Good** | **0.335** |
| | PERSIANN-CCS | 0.80 | 0.74–0.85 | 0.60 | 0.14 | Poor | $3.47 \times 10^{-4}$ |
| | **PERSIANN** | **0.82** | **0.78–0.87** | **0.64** | **0.38** | **Moderate** | **0.504** |
| | **GPM-3IMERGMv6** | **0.88** | **0.84–0.91** | **0.76** | **0.55** | **Good** | **0.154** |
| | **GPM-3IMERGDLv6** | **0.87** | **0.83–0.91** | **0.74** | **0.52** | **Good** | **0.291** |
| October | CHIRPS-2.0 | 0.79 | 0.75–0.83 | 0.58 | 0.25 | Moderate | 0.028 |
| | PERSIANN-CDR | 0.75 | 0.71–0.78 | 0.5 | 0.10 | Poor | 0.046 |
| | PERSIANN-CCS | 0.77 | 0.72–0.82 | 0.54 | 0.12 | Poor | 0.039 |
| | PERSIANN | 0.75 | 0.70–0.79 | 0.5 | 0.04 | Poor | 0.027 |
| | GPM-3IMERGMv6 | 0.77 | 0.73–0.82 | 0.54 | 0.18 | Poor | 0.004 |
| | GPM-3IMERGDLv6 | 0.81 | 0.77–0.85 | 0.62 | 0.19 | Poor | 0.011 |
| November | **CHIRPS-2.0** | **0.82** | **0.78–0.85** | **0.64** | **0.33** | **Moderate** | **0.594** |
| | **PERSIANN-CDR** | **0.81** | **0.77–0.84** | **0.62** | **0.27** | **Moderate** | **0.430** |
| | **PERSIANN-CCS** | **0.81** | **0.76–0.85** | **0.62** | **0.33** | **Moderate** | **0.895** |
| | PERSIANN | 0.76 | 0.72–0.81 | 0.52 | 0.13 | Poor | 0.037 |
| | **GPM-3IMERGMv6** | **0.80** | **0.75–0.84** | **0.6** | **0.29** | **Moderate** | **0.999** |
| | **GPM-3IMERGDLv6** | **0.81** | **0.77–0.85** | **0.62** | **0.33** | **Moderate** | **0.900** |
| December | **CHIRPS-2.0** | **0.82** | **0.79–0.86** | **0.64** | **0.35** | **Moderate** | **0.278** |
| | **PERSIANN-CDR** | **0.81** | **0.78–0.84** | **0.62** | **0.31** | **Moderate** | **0.617** |
| | **PERSIANN-CCS** | **0.78** | **0.73–0.83** | **0.56** | **0.25** | **Moderate** | **0.456** |
| | PERSIANN | 0.76 | 0.71–0.80 | 0.52 | 0.18 | Poor | 0.022 |
| | **GPM-3IMERGMv6** | **0.83** | **0.79–0.87** | **0.66** | **0.38** | **Moderate** | **0.999** |
| | GPM-3IMERGDLv6 | 0.77 | 0.71–0.81 | 0.54 | 0.13 | Poor | $8.22 \times 10^{-4}$ |

Results in bold indicate adequate product performance for drought detection.

The results in Table 5 show that a single product to estimate the rain by remote sensing, used to calculate SPI, could not detect droughts in the region for all months of the year, requiring the use of other products to characterize better the drought regime in the region. Drought estimation with rain estimated by PERSIANN-CCS presented good performance for detecting droughts in April. CHIRPS-2.0 was the precipitation product that resulted in the best performance in January, February, March, May, September, and November. The product GPM-3IMERGMv6 presented a moderate performance for detecting droughts in December.

The study in [22] concluded that the PERSIANN-CDR and CHIRPS data were suitable for monitoring droughts with SPI in eastern mainland China. However, the results for western China indicated their inadequacy for monitoring droughts. In the case of the north–central region of the State of Mato Grosso, these products were effective in detecting droughts in January, February, March, May, September, November, and December, but only CHIRPS was valid for January and May (Table 5). The study in [47] used CHIRPS data to monitor drought and humidity throughout the semi-arid region of Midwest Argentina. According to these authors, CHIRPS is an adequate tool to monitor precipitation anomalies for periods longer than one month. The results obtained in the present study diverge from that research since the CHIRPS data were used for monthly drought monitoring.

The $H_0$ hypothesis of the McNemar test was rejected for all products in October indicating a significant difference between the drought detection with data from rain gauge stations and products derived from remote sensing. Therefore, drought monitoring using satellite information is not recommended during this month.

The study in [48] found that the accuracy of rainfall data estimated by the TRMM depends on several factors such as region, the season of the year, time and space scales. These authors also presented the results of other studies that showed the variability of the TRMM accuracy for different regions of the world as well as for the intensity and other characteristics of the rain. The results presented in Table 5 indicate that this variability in the accuracy of the orbital remote sense rainfall measurement is also valid for the CHIRPS, PERSIANN, and GPM products tested in this work and detailed in Table 2.

The causes for this variability are related to different factors such as seasonality, type of rain, the influence of synoptic systems, and surface factors such as soil occupation/roughness. Because of this, further studies should be carried out in the region to identify the intervening factors in the rain performance estimated by remote sensing and in the drought estimate with the SPI.

The results show that the rain deficit in the study region can be monitored with remote sensing during practically the entire soybean development period, that is, from October/November to February/March. Moreover, soybean is the preferred crop for producers, showing the highest commercial value other than corn.

Figure 5 shows the drought maps for the period from November 2015 to March 2016, obtained from the remote sensing products validated in Table 5. These maps corroborate with those obtained with rain gauge data measured on the surface (Figure 3), but with higher detailing regarding the spatial variability of the drought event in the region.

Most of the region in November 2015 had no drought, but some municipalities such as Cláudia and Santa Rita do Trivelato had most of their area affected by a rain deficit. Almost the entire north–central region of the State of Mato Grosso was affected by drought in December and February. Moreover, January and March presented no rain below normal.

As previously discussed, the drought that occurred in certain locations in November and almost the entire region in December and February was the main factor responsible for the soybean crop shortfall in the 2015/2016 growing season.

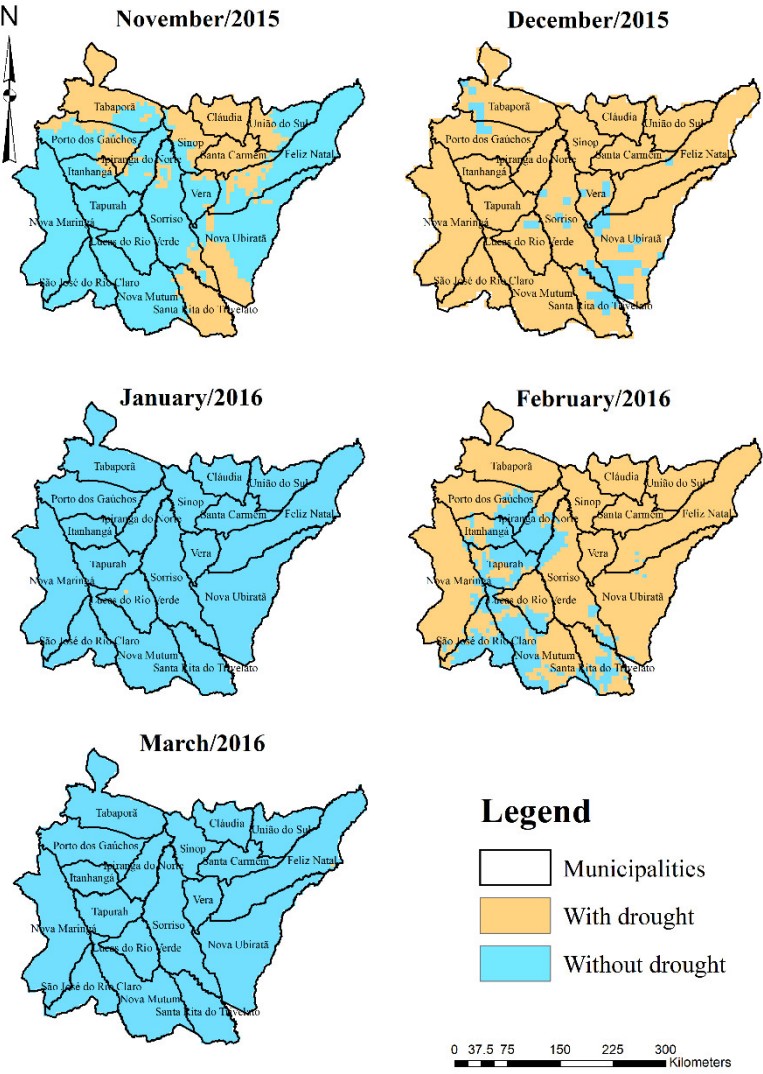

**Figure 5.** Drought maps resulting from the calculation of SPI-1 with CHIRPS-2.0 (November, January, February and March) and GPM-3IMERGMv6 data (December) for the period from November 2015 to March 2016 in the north–central region of the State of Mato Grosso, Brazil.

Researchers from the Empresa Brasileira de Pesquisa Agropecuária (EMBRAPA) made a technical report in March 2016 on the situation of the 2015/2016 growing season in the study region requested by the Associação Brasileira de Produtores de Soja (Aprosoja). Details of this report are available in [49]. This report confirms the results obtained in this research, as it associates a reduction in soybean production with the water deficits that occurred from 30 November to 24 December, 2015, and February 2016. The researchers highlighted that the periods with low precipitation coincided with the vegetative development and flowering and grain-filling stages of many soybean cultivations, resulting in a low development and abortion of flowers and empty pods, respectively.

In addition, [49] associated the soybean production losses with an increase in the incidence of plants with black root rot caused by *Macrophomina phaseolina*. According to the authors, the occurrence of a drought period followed by a humidity period favored the root invasion by the fungus, leading to wilt and deterioration of the woody root tissue as it progresses. Figure 5 shows the alternation of dry and wet periods, mainly between December 2015 and March 2016, which may have favored the increased incidence of *Macrophomina phaseolina* on soybean plants.

The analyses showed the possibility of monitoring droughts with SPI calculated through historical rain series derived from remote sensing. This approach, even not characterizing the

drought severity, allows a synoptic analysis of this natural disaster and can optimize the direction of measures to mitigate impacts.

The density of rain gauge stations in the Amazon region, where the study area is located, is very low compared to other regions of Brazil. The promising results obtained in this study for monitoring droughts with indirect rain measurements originating from remote sensing validate an important resource to deal with the limitations of the hydrometeorological information collection network in the region.

According to [50], the estimation of rain with a high spatial and temporal resolution by remote sensing (radar) can be a promising approach to mitigate the uncertainty resulting from the high spatial variability of precipitation in an area, particularly in combination with surface data (rain gauges). The results obtained for the north–central region of the State of Mato Grosso (Table 3 and Figure 5) prove that this statement is also valid for the drought estimation using orbital remote sensing products duly validated with surface data.

A comparative analysis of the results obtained in this study with those recently published show that drought indices and indirect satellite rain measurements work specifically for different world regions, as well as their correlation with the agricultural production. In this sense, studies that use procedures similar to those employed in this research and that validate a given index regarding its ability to characterize the drought of a region and its impact on the agricultural sector are essential and strategic.

The use of soil moisture estimated by remote sensing also can be used in further research to improve the drought monitoring in the study region. Works such as [51,52] evaluated different estimates of soil moisture derived from remote sensing and found that these products have adequate performance for this purpose. The soil moisture and other physical characteristics of the soil would make detection of droughts and their impacts on crops more robust. In this work, we do not use the soil moisture estimated by satellite due to the absence of a soil database for the validation of satellite products, such as the soil moisture and the water storage capacity in the soil measured in-situ. These are not only a limitation of the study region but many in developing countries. There are works aimed to improve the soil database conducted by the association of soy producers in partnership with universities in the northern middle region of Mato Grosso, which will allow future work to validated soil moisture products to monitor drought.

## 4. Conclusions

The obtained results allow us to conclude that:

- The standardized precipitation index (SPI) is effective in characterizing and monitoring the meteorological drought in the north–central region of the State of Mato Grosso. There is a relationship between the drought events detected by SPI and the reduction in grain production that occurs under rainfed conditions.
- The SPI calculated from historical rain series estimated by remote sensing allows monitoring droughts. This approach allows for a higher detail of the spatial variability of droughts in the Cerrado–Amazon transition region located in the north–central region of the State of Mato Grosso, especially during the soybean development period.
- The indirect measures of precipitation from remote sensing associated with SPI on the monthly scale (SPI-1) that best perform for detecting droughts in the study region are PERSIANN-CCS (April), CHIRPS-2.0 (January, February, March, May, September and November), and GPM-3IMERGMv6 (December).

**Author Contributions:** Conceptualization, M.Â.C.C.d.C. and E.M.U.; data curation, M.Â.C.C.d.C. and E.M.U.; methodology, M.Â.C.C.d.C. and E.M.U.; software, M.Â.C.C.d.C. and E.M.U.; Supervision, M.Â.C.C.d.C. and E.M.U.; validation, M.Â.C.C.d.C., E.M.U. and D.D.d.S.; writing—original draft preparation, M.Â.C.C.d.C. and E.M.U.; writing—review and editing, D.D.d.S, U.R.V.A., C.A.d.S.M., M.F.d.S.J., I.F.d.C. and M.A.d.S.A.M. All authors have read and agreed to the published version of the manuscript.

**Funding:** This study was partially funded by the Conselho Nacional de Desenvolvimento Científico e Tecnológico (CNPq), the Coordenação de Aperfeiçoamento de Pessoal de Nível Superior—Brazil (CAPES)—Financing Code 001, and Programa de Apoio à Publicação em Periódicos Qualificados (PROPeq/PROPG-UFMT), process SEI n° 23108.082146/2020-14.

**Conflicts of Interest:** The authors declare no conflict of interest.

## Appendix A

**Table A1.** Information related to the rain gauge stations used in the study.

| ID | Code | Name | Latitude (°) | Longitude (°) | Available Data |
|----|------|------|--------------|---------------|----------------|
| 1 | 01052000 | Vila São José Xingu | −10.81 | −52.75 | 1976–2017 |
| 2 | 01053001 | Fazenda Santa Emília | −10.54 | −53.61 | 1976–2017 |
| 3 | 01055003 | Fazenda Tratex | −10.96 | −55.55 | 1994–2017 |
| 4 | 01056001 | Estância Buriti | −10.40 | −56.42 | 2000–2017 |
| 5 | 01154000 | Rancho de Deus | −11.00 | −54.81 | 1985–2017 |
| 6 | 01154001 | Santa Felicidade | −11.93 | −55.00 | 1982–2017 |
| 7 | 01154002 | Fazenda Rio Negro | −11.52 | −54.36 | 1999–2017 |
| 8 | 01154004 | Cláudia | −11.49 | −54.87 | 2004–2017 |
| 9 | 01154005 | Riacho de Deus | −11.13 | −54.48 | 2006–2017 |
| 10 | 01155000 | Cachoeirão | −11.65 | −55.70 | 1975–2017 |
| 11 | 01156000 | Fazenda Itauba | −11.47 | −56.43 | 1982–2017 |
| 12 | 01156001 | Sinop (Fazenda Sempre Verde) | −11.69 | −56.45 | 1983–2017 |
| 13 | 01156002 | Tabaporã | −11.30 | −56.83 | 2004–2017 |
| 14 | 01156003 | Nova Americana | −11.64 | −56.16 | 2004–2017 |
| 15 | 01157000 | Porto dos Gaúchos | −11.54 | −57.42 | 1973–2017 |
| 16 | 01157001 | Juara | −11.25 | −57.51 | 1983–2017 |
| 17 | 01157002 | Olho D'água | −11.77 | −57.04 | 1999–2017 |
| 18 | 01254001 | Agrovensa | −12.81 | −54.75 | 1982–2017 |
| 19 | 01254002 | Consul | −12.37 | −54.49 | 1997–2012 |
| 20 | 01254003 | Agropecuária Três Irmãos | −12.80 | −54.25 | 2000–2017 |
| 21 | 01255001 | Teles Pires | −12.68 | −55.79 | 1976–2017 |
| 22 | 01255002 | Núcleo Colonial Rio Ferro | −12.52 | −54.91 | 1976–2017 |
| 23 | 01256002 | Fazenda Divisão | −12.98 | −56.18 | 1999–2017 |
| 24 | 01257000 | Brasnorte | −12.12 | −58.00 | 1984–2017 |
| 25 | 01354000 | Fazenda Agrochapada | −13.45 | −54.28 | 1976–2017 |
| 26 | 01354001 | Agropecuária MALP | −13.34 | −54.08 | 1999–2017 |
| 27 | 01354002 | Fazenda Itaguaçu | −13.14 | −54.44 | 2003–2017 |
| 28 | 01355001 | Porto Roncador | −13.56 | −55.33 | 1985–2017 |
| 29 | 01356002 | Nova Mutum | −13.82 | −56.12 | 1985–2017 |
| 30 | 01356004 | São José do Rio Claro | −13.45 | −56.73 | 2004–2017 |
| 31 | 01357000 | Nova Maringá | −13.03 | −57.09 | 1982–2017 |
| 32 | 01357001 | Campo Novo do Parecis | −13.69 | −57.89 | 2000–2017 |
| 33 | 01455009 | Fazenda Rio Novo | −14.22 | −55.51 | 2000–2017 |
| 34 | 01457003 | Deciolândia | −14.18 | −57.51 | 1982–2017 |

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
