# Peer review of "Drought Monitoring Based on Remote Sensing in a Grain-Producing Region in the Cerrado–Amazon Transition, Brazil"

_water, doi:10.3390/w12123366_

Round 1
Reviewer 1 Report
The manuscript corresponds to the direction of the special issue and is recommended for publication. Wish: to continue research using remote sensing proposed in the article, which takes into account meteorological drought processes.
Author Response
Dear reviewer,
We would like to thank you for your consideration and suggestions of our manuscript “Drought monitoring based on remote sensing in a grain-producing region in the cerrado-amazon transition, Brazil”. Your consideration is sincerely appreciated.
We have addressed each of the reviewer's comments and modified our paper accordingly. All changes in the manuscript are highlighted in red. The following pages contain the authors’ feedback. Reviewer comments are shown in black, author responses are shown in red in the attached file.

Reviewer 2 Report
The manuscript tries to evaluate the SPI drought index using various ground and satellite measurements over Cerrado-Amazon transition region, Brazil. The manuscript is overall well-prepared. I have several concerns regarding this manuscript:
- Introduction section needs improved. There are many basic definition/introduction about drought which I think is not necessary for academic publication in Water. Moreover, the authors fail to highlight the novelty or importance of their work.
- It’s unclear to me that how SPI from remotely-sensed and ground precipitation are compared. Dose the authors compare SPI of specific pixel that have rain gauges in it? Methods for calculating metrics in Table 6 should be provided. They are more important than Eq. 1-9.
- The conclusion is interesting. Performance of different satellite precipitation depends on time. I suggest the authors to compare the satellite precipitation directly with ground data to look into the reason why a given dataset is only suitable for a specific month.
- Table 1 can be moved to appendix
Author Response

(The authors gave the same response as above.)

Reviewer 3 Report
The paper titled “Drought monitoring based on remote sensing in a grain-producing region in the cerrado-amazon transition, Brazil” evaluated the skill of a frequently-used drought index, i.e., the standardized precipitation index (SPI) in monitoring of meteorological drought in the Cerrado-Amazon transition region, Brazil. The capability of six satellite-based precipitation products in detecting the drought was also examined by using 34 rain gauge stations. The results indicate the SPI is effective in monitoring the meteorological drought in the study areas, and the satellite precipitation products show different skills in different months.
Though the topic in interesting, the motivation and novelty of this paper are not well expounded and presented. The SPI along with the satellite precipitation data have been extensively used to detect drought in numerous previous studies. So what is the new finding in this paper and what is the problem solved in this study? This was not well answered in the manuscript. Particularly in the introduction part, the authors just stated the work of other researchers, but there is no summary of the existing problems which the authors intend to solve (or study). The manuscript is also not well written and there are too many paragraphs in the current paper. I believe the authors have to do more work to further clarity the motivation and novelty of their paper before a full publication is warranted.
Other specific comments:
1) Line 86: “PHDI and KDBI hydrological drought indices”? Rephrase.
2) Line 147: clarify how did you calculate the probable average monthly precipitation.
3) Line 168 and 169: do not list the full name of SPI and WMO again since they have been introduced in former content.
4) Line 172: the SPI can be calculated at multi-time scales, clarify why you only calculate SPI-1 in your study.
5) Line 235 to 237: why? Did you validate the accuracy of the satellite precipitation data by using the rain gauge measurements?
6) Figure 4: why the production in 15/16 is higher than that of 12/13 since there is no drought in 12/13 (see Figure 3).
7) Line 319-322: why?
Author Response
Dear reviewer,
We would like to thank you for your consideration and suggestions of our manuscript “Drought monitoring based on remote sensing in a grain-producing region in the cerrado-amazon transition, Brazil”. Your consideration is sincerely appreciated.
We have addressed each of the reviewer's comments and modified our paper accordingly. All changes in the manuscript are highlighted in red. The following pages contain the authors’ feedback. Reviewer comments are shown in black, author responses are shown in red.

Round 2
Reviewer 2 Report
The authors have addresses some of my concerns. However, my concern regarding why performance of different satellite precipitation depends on time is not addressed. This is important, because the authors claim that the contribution of this manuscript is to evaluate satellite precipitation in detecting drought. Actually, there already many publications trying to address this problem. Therefore, the authors need to find out the reason why certain product is only suitable for some specific month so the manuscript can be more meaningful.
Author Response
Dear reviewer,
We would like to thank you for your consideration and suggestions of our manuscript “Drought monitoring based on remote sensing in a grain-producing region in the cerrado-amazon transition, Brazil”. Your considerations are sincerely appreciated.
We have addressed each of the reviewer's comments and modified our paper accordingly. All changes in the manuscript are highlighted in red. The following pages of the attached file contain authors’ feedback. Reviewer comments are shown in black, author responses are shown in red in the attached file.

Reviewer 3 Report
The authors have addressed most of my previous comments and the paper has been improved. But I believe more discussion is needed before publication.
1) The authors mainly investigated the relationship between the drought events detected by SPI and the grain production. I believe the grain production is more related to the agricultural drought than meteorological drought, and soil moisture is the most important parameter for agricultural drought. Nowadays, many remotely sensed soil moisture products are freely available (see Zeng et al., 2015, http://dx.doi.org/10.1016/j.rse.2015.03.008; Cui et al., 2018, doi:10.3390/rs10010033). It doesn’t know why the authors adopted precipitation rather than soil moisture to detect the relationship between drought and grain production. This issue should be explained and discussed in the manuscript.
2) The authors stated in the response "As the predominant cultures in the region are temporary, we believe that the adoption of periods longer than one month for the SPI was not compatible.". What do you mean? Please give more details about your explanation.
3) I wonder if you validate the accuracy of the satellite precipitation data by using the rain gauge measurements directly (not by the drought estimates)? Are these results consistent with the performance of the satellite products shown in Table 5?
4) The authors stated in the response "There was an increase in production of 15/16 even with drought greater than the 12/13 due to an increase in planted area. The study region is located in the north and central west of Brazil, where the exploitation of land by primary production is still expanding, with the removal of vegetation cover for the implementation of agriculture.” Please clarify this issue in the manuscript.
5) I suggest that the author reduce the number of paragraphs (some paragraphs can be combined into one) to make the paper more readable.
Author Response
Dear reviewer,
We would like to thank you for your consideration and suggestions of our manuscript “Drought monitoring based on remote sensing in a grain-producing region in the cerrado-amazon transition, Brazil”. Your considerations are sincerely appreciated.
We have addressed each of the reviewer's comments and modified our paper accordingly. All changes on the manuscript are highlighted in red. The following pages of the attached file contain authors’ feedback. Reviewer comments are shown in black, author responses are shown in red in the attached file.

Round 3
Reviewer 2 Report
I'm satisfied with the responses from authors.
Author Response
Dear reviewer,
We would like to thank you for your consideration and suggestions of our manuscript “Drought monitoring based on remote sensing in a grain-producing region in the cerrado-amazon transition, Brazil”. Your suggestions were fundamental for the improvement of the work and allowed us to reflect on the results presented and the methodology used, which will be of great help for our future work.
Sincerely,
Uilson Ricardo Venâncio Aires
Reviewer 3 Report
The authors addressed some of my concerns but also sometimes misunderstood my points. For some comments which I have pointed out in the review report, they only explained in the response but did not make any revision or discussion in the manuscript. The following issues should be carefully addressed before publication in Water.
1) It is not my intention to ask the authors do other experiments by using satellite soil moisture products. However, the authors can discuss the importance of adding (or using) satellite soil moisture products for drought monitoring, particularly the agriculture drought (such as the case in the study) monitoring in the manuscript (for instance, you can add such discussion for your future work). It is also important to let the readers know that there are a number of satellite soil moisture products freely available (such as the references I provided for the authors) which can be used for their applications.
2) Please clarify the reasons you only calculated SPI-1 in your study in the manuscript (not just in the response), to make this issue clear to the readers.
Author Response

(The authors gave the same response as above.)
